# Spontaneous Remission of Primary Aldosteronism with Mineralocorticoid Receptor Antagonist Therapy: A Review

**DOI:** 10.3390/ijms232213821

**Published:** 2022-11-10

**Authors:** Xurong Mai, Mitsuhiro Kometani, Takashi Yoneda

**Affiliations:** Department of Health Promotion and Medicine of the Future, Graduate School of Medical Sciences, Kanazawa University, Kanazawa 920-8641, Ishikawa, Japan

**Keywords:** primary aldosteronism, remission, mineralocorticoid receptor antagonist, steroidogenesis enzyme

## Abstract

In this review, we describe previous basic and clinical studies on autonomous aldosterone production. Over the past decades, mineralocorticoid receptor antagonists (MRAs) have been found to concentration-dependently inhibit steroidogenesis in different degrees. However, many studies have proven the suppressive effects of MRAs on the activities of hormone synthase. The probable factors of cytochrome P-450 reduction, both in microsomes and mitochondria, have also been considered: (1) one of the spironolactone metabolite forms had destructive function, except canrenone, (2) 7α-thio-spironolactone was an obligatory intermediate in the spironolactone-induced CYP450 decrease, and (3) the contributing steroids should have 7α-methylthio or 7α-methylsulfone groups. In previous clinical research, spironolactone-body-containing cells showed a type II pattern of enzyme activity (i.e., enhanced 3β-hydroxysteroid dehydrogenase, glucose-6-phosphate, and NADP-isocitrate dehydrogenase activities and weaken succinate dehydrogenase activity), and the subcapsular micronodules composed of spironolactone-body-containing cells also exhibited a type II pattern and excess aldosterone secretion, indicating that the subcapsular micronodules might be the root of aldosterone-producing adenoma. Moreover, combined with the potential impeditive function to aldosterone secretion, a few cases of spontaneous remission of primary aldosteronism, with normal ranges of blood pressure, plasma potassium, plasma renin activity, and aldosterone renin ratio, have been reported after long-term treatment with MRAs.

## 1. Introduction

Primary aldosteronism (PA) is a leading form of secondary hypertension, a disease that presents with hypertension due to autonomous aldosterone hypersecretion from the adrenal glands. Compared to patients with essential hypertension (EHT), PA is associated with a three- to four-fold higher frequency of cardiovascular complications [1] and has a poor prognosis, requiring accurate diagnosis and qualified treatment. Furthermore, it is estimated that there are at least two million patients with PA in Japan, many of whom are misidentified as having EHT and are inappropriately treated [2,3]. PA can be divided into two subtypes, unilateral and bilateral lesions, depending on the site of lesion. The Endocrine Society and Japan Endocrine Society guidelines [4,5] recommend laparoscopic adrenalectomy on the affected side as the standard treatment for unilateral lesions. However, for bilateral PA (BPA), drug therapy with mineralocorticoid receptor antagonists (MRAs), such as spironolactone (SL) and eplerenone (EP), are recommended. Usually, drug therapy aims to exert an antihypertensive effect by suppressing the function of aldosterone. Remission of autonomous aldosterone production by MRAs in PA cases has been reported. In this review, we describe previous basic and clinical studies on autonomous aldosterone production. 

## 2. Inhibition of Steroid Hormone Synthesis by Mineralocorticoid Receptor Antagonists

SL has been widely used as an MRA for different types of hypertension since its discovery in the mid-1950s. SL is used for the oral treatment of PA as an MRA. SL has been reported to inhibit aldosterone synthesis in the adrenal cortex and aldosterone antagonism in the periphery. Inhibition of hormone synthesis has been reported for 11β-hydroxylase [6,7,8,9], 21-hydroxylase [8,9,10,11], 18-hydroxylase [6,7], and 17α-hydroxylase [10,11,12,13] (Table 1). The inhibitory effect of SL is thought to be due to the disruption of cytochrome P450 in steroid-synthesizing cells [9]. Menard et al. [14] reported that SL disrupted cytochrome P450 in adrenal and testicular microsomes, resulting in enzyme inhibition.

### 2.1. Impeditive Effect on Aldosterone Steroidogenesis

By comparing the effects of SL, canrenone, and potassium canrenoate on cytochrome P450 in human and bovine adrenal glands, Cheng et al. [6] found that all three agents reduced the rate of 11-deoxycorticosterone (DOC) to corticosterone and corticosterone to 18-hydroxy-corticosterone, although none of them interfered with CYP450 in mitochondria by themselves. Moreover, the potency and inhibition of aldosterone synthesis were found to be related to their affinity for cytochrome P450. This result is the same as that reported by Abshagen et al. [7]. A considerable increase in plasma DOC was detected after SL administration (100 mg three times a day) in five healthy male volunteers, suggesting inhibition of 11β- and 18-hydroxylase. In Menard et al.’s study [10], administration of SL in vivo at doses of 100 mg/kg caused a remarkable reduction in cytochrome P-450 in the adrenal glands and testes, 60% to 70% and 40% to 50%, respectively, indicating that steroids containing 7α-methylthio or 7α-methylsulfone groups could be the main reason for this destruction, because the destruction did not occur in the groups lacking these steroids. At the same time, through the rate of progesterone to 17α-hydroxyprogesterone and 21-hydroxyprogesterone, the activities of adrenal 17α-hydroxylase and 21-hydroxylase were detected, with a 50% to 80% loss, similar to the heme concentration of cytochrome P-450. Similarly, Colby’s study [8] showed that both SL and canrenone could concentration-dependently inhibit microsomal 21-hydroxylation and mitochondrial 11β-hydroxylation, although the effect of canrenone was better when using mitochondria and microsomes derived from guinea pig adrenal glands. 

In addition, another report focusing on aldosterone synthesis influenced by different MRAs, SL, prorenone, SC 19886, SC 26304, and SC 27169, by Netchitailo et al. [15] revealed that SL could inhibit 75% of aldosterone synthesis, but it would recover partially after drug infusion in frog adrenal glands. Furthermore, an experiment conducted by Ye et al. used human adrenocortical H295R cells [16]. After comparing the effects of SL and EP, there were several significant consequences: (1) SL could inhibit Ang II-stimulated aldosterone production by 80% and Ang II-stimulated cortisol production by 74%; (2) SL could inhibit pregnenolone metabolism to both aldosterone and cortisol by 67% and 74%, respectively; and (3) EP had no effect on basal, Ang II, or forskolin stimulation of aldosterone or cortisol production.

According to these studies, canrenone, one of the major active forms of SL, is often used to compare its effects with SL. SL is a prodrug with a short half-life (1.4 h), and the long half-life of canrenone (16.5 h) has also led to the conclusion that canrenone might be the main therapeutic form of SL treatment [6]. This can be proven in some way through the better inhibition of enzyme activity by canrenone and the inhibition of aldosterone synthesis in different human and animal experiments. 

### 2.2. Peripheral Influence on Cortisol Secretion

In addition to aldosterone secretion from the zona glomerulosa, the production of cortisol from the zona fasciculata has begun to attract attention. A study designed by Rourke and Colby [17] used adrenal glands collected from guinea pigs to probe the inhibition of cortisol production by SL. Both SL and its intermediate, 7α-thio-SL, which is also one of the three major active forms of SL, decreased cortisol production in a time- and concentration-dependent manner by nearly 50%, with 7α-thio-SL being far more potent. At the same time, Rourke et al. [12] focused on the mechanism of action of SL on cortisol production. In their experiment, they incubated adrenocortical cells derived from guinea pigs with three different agents: (1) the 17α-hydroxylase inhibitor SU-10′603, (2) 11β-hydroxylase inhibitor metyrapone, (3) cholesterol sidechain cleavage inhibitor aminoglutethimide, and SL or 7α-thio-SL. Eventually, they realized that the direct effects of SL and 7α-thio-SL on cortisol production resulted from the selective inhibition of 17α-hydroxylation, whereas metyrapone and aminoglutethimide did not change the efficacy of SL or 7α-thio-SL on cortisol production.

Through further studies, we can clearly see that SL has the ability to reduce the production of aldosterone and cortisol in several ways. On the one hand, the enzyme activities of 21-hydroxylase, 11β-hydroxylase, and 18-hydroxylase will be affected by SL, leading to the reduction in aldosterone; on the other hand, the selective inhibition of 17α-hydroxylase will certainly decrease the production of cortisol. SL, canrenone, and 7α-thio-SL were potent inhibitors of aldosterone synthesis and cortisol production. 

### 2.3. Interrelation with Cytochrome P-450

The in vitro experiment by Menard et al. [10] showed that in the presence of NADPH, incubation of testicular microsomes with SL caused a sharp destruction of >80% of the amount of cytochrome P-450, but the decrease was not obvious without NADPH in adrenal and testis tissues. The studies performed in this study, in vivo and in vitro, indicated that SL could reduce the concentration of cytochrome P-450 in the adrenal gland or testes, not directly by itself, but probably in one of its metabolite forms instead. Greiner et al. [9] similarly reported that the incubation of adrenal microsomes with SL plus NADPH resulted in the decline of cytochrome P-450, but this phenomenon did not occur in the metabolite of SL, canrenone, with or without NADPH, considering that canrenone was neither a reactive metabolite nor could it be mediated by adrenal microsomes, although both agents were responsible for inhibiting the binding of steroid substrates to cytochrome P-450 and reducing the activities of adrenal mitochondria (11β-hydroxylase) and microsomes (21-hydroxylase). However, a subsequent study by Kossor et al. [13] implied that 7α-thio-SL was an obligatory intermediate in the SL-induced CYP450 decrease. The initial step of SL activation is the deacetylation of SL to 7α-thio-SL, but the subsequent pathways require the destruction of CYP450. Incubation of guinea pig adrenal microsomes with 7α-thio-SL plus NAPDH resulted in a significant decline in CYP450 and 17α-hydroxylase by >50%. In contrast, after adding antisera to 17α-hydroxylase, no change in CYP450 degradation occurred, which showed that the activation of 7α-thio-SL required 17α-hydroxylase. 

Kossor and Colby [11] used a high dose (100 mg/kg) and low dose (25 mg/kg) of SL to determine dose-dependent effects on the outer (zona glomerulosa plus zona fasciculata) and inner (zona reticularis) zones in guinea pigs. The results showed that high doses of SL significantly decreased 17α-hydroxylase and 21-hydroxylase, accompanied by the degradation of mitochondrial P450 and heme concentrations. However, for low doses of SL, microsomal P450 activities were inhibited, whereas 21-hydroxylase declined in the inner zone only, and the amount of mitochondrial P450 did not change in either zone. Additionally, a high dose of SL altered the gross appearance of the adrenal glands, which has also been found with low doses of SL, indicating that a high dose of SL may have a number of unclear, or even toxic, effects on adrenal glands. A high dose of SL (100 mg/kg) is widely used in experiments, and Menard et al. [14] used this dose of SL in male dogs. 

Interestingly, they found that a high dose of SL could decrease the concentrations of testosterone in testicular and peripheral venous plasma, and the concentration of cortisol in adrenal venous plasma, by 60–75% and 50–65%, respectively. In contrast, canrenone and potassium canrenoate did not cause any significant changes. Although some of the positive results were meaningful and helpful for the study of SL, the difficulty between practical usage and experimental results is worth noting. Ye et al. [16] reported that the inhibitory effects of SL occurred at concentrations far higher than those needed to block mineralocorticoid receptors: up to 400 mg per day.

## 3. Analysis of Spontaneous Remission of Primary Aldosteronism after Long-Term Treatment with Mineralocorticoid Receptor Antagonists

MRAs, such as canrenone or potassium canrenoate, SL, and EP, play an essential role in hypertension because of their function in reducing blood pressure (BP) and improving hypokalemia. However, with further studies on MRAs (Table 2), especially SL, several studies have shown that the characteristics of SL-body-containing cells could be used to determine the cause of PA [18], as well as find the probable root of aldosterone-producing-adenoma (APA), due to the subcapsular micronodules containing these cells, derived from adrenals, with excess aldosterone secretion [19]. In the case of an inhibitory effect on aldosterone secretion [20], long-term treatment with SL might lead to spontaneous remission of PA [21,22,23,24,25].

### 3.1. Biochemical Characteristics of Spironolactone-Body-Containing Cells

Aiba et al. [18] focused on the SL body, a formation induced in aldosterone-producing cells of the human adrenal cortex after SL administration, detected in aldosterone-producing adenomas, adrenal tissues attached to APA, and bilateral adrenal hyperplasia tissues. In their study, SL-body-containing cells and their neighboring cells showed a strengthening function of 3β-hydroxysteroid dehydrogenase (3β-HSD), the only enzyme out of CYP450 in the adrenal pathway of corticosteroid synthesis. Moreover, the S-body-containing cells also showed improved glucose-6-phosphate (G6PD) and NADP-dependent isocitrate dehydrogenase (NADP-ICDH) activities, and decreased succinate dehydrogenase (SDH) activity, which is known as the type I pattern of enzyme activity. The majority of adrenal cells without S bodies, however, showed an opposite pattern of enzyme activity, known as the type II pattern, with weak 3β-HSD, G6PD, and NADP-ICDH activities and intense SDH activity. Interestingly, all zona glomerulosa cells in a PA patient with bilateral adrenal hyperplasia exhibited enhanced 3β-HSD activity, irrespective of the presence or absence of SL bodies. In summary, this characteristic of SL could be used to distinguish the causes of PA, cortical adenoma, or bilateral adrenal hyperplasia. Furthermore, Shigematsu et al. [19] found subcapsular micronodules composed of SL-body-containing cells with intense expressions of 3β-HSD, 11β-OH, 18-OH, and 21-OH, but not 17α-OH. After comparing steroid enzyme activities from UAH, BAH, and adrenal cortices adhering to APA, the nodules observed in UAH and the adrenal cortices adhering to APA in that study also exhibited enzyme activities similar to the type I pattern. Although hyperplastic zona glomerulosa and nodules generally showed decreased steroidogenic activities in the adrenal tissues attached to APA, these nodules with prominent aldosterone production were found at a high frequency, indicating that the subcapsular micronodules might be the root of APA. 

### 3.2. Potential Function in Aldosterone Secretion

Generally, during MRA treatment, plasma aldosterone concentration (PAC) is expected to concomitantly increase in patients with PA. We [20] found that SL and EP could raise PAC (135 ± 57 to 213 ± 90 pg/mL) in PA patients (*n* = 54). Similarly, a clinical study by Fourkiotis et al. [21], which compared the efficacies of SL and EP for PA patients (*n* = 29), corroborated this conclusion by revealing a PAC increase from 58.3 ± 9.0 to 159.3 ± 35.9 ng/L. However, there are also some interesting cases reporting the decrease in PAC during treatment. A 51-year-old female patient with PA and a 3-year history of hypertension showed improvements in the PAC and aldosterone secretion after two periods of SL treatment [22]. The first treatment was 100 mg/day of SL for 1 week and 200 mg/day for the next three weeks. Treatment was terminated with a reasonable decrease in aldosterone: >100 pg/mL in contrast to the initial concentration of approximately 210 pg/mL. After a 7-week cessation of any medication, the aldosterone level increased to 120 pg/mL and the aldosterone secretion rate, measured by the urine method, was 352 μg/24 h. During the second period of SL treatment, the patient received 200 mg/day and 400 mg/day in the first and later weeks, respectively. As a result, four months later, the aldosterone secretion rate (urine method) reduced remarkably to 68 μg/24 h, while the aldosterone level remained within a range of 30–70 pg/mL, suggesting that SL might have an ability to inhibit aldosterone secretion. In addition, the regulation of PAC is associated with not only the medical treatments but also the metabolic syndrome, including the waist circumference in men [23], high-density lipoprotein [24], and salt intake [25]. The improvement in these factors will significantly decrease PAC, and we should also pay more attention to them during MRA treatment.

### 3.3. Spontaneous Remission of Primary Aldosteronism after Long-Term Treatment with Mineralocorticoid Receptor Antagonists

Armanini et al. [26] recruited 15 patients (nine male and six female patients) with PA who had been treated with potassium canrenoate (25–100 mg/day) for at least 2.5 years. After a one-month withdrawal of potassium canrenoate, they found that the decrease in the aldosterone/plasma renin activity (PRA) ratio in the upright position had a significant inverse relationship with the duration of potassium canrenoate treatment (*p* = 0.045). Moreover, they claimed that 12 of 15 patients had normal aldosterone/PRA ratios, which might be due to the repaired sensitivity of the adrenal gland to Ang II caused by long-term therapy with aldosterone receptor blockers. This hypothesis was further supported by a subsequent study performed by the authors [27], which focused on three patients with PA who had not taken potassium carenoate as a treatment for >5 years. 

In addition to potassium carenoate, some research focusing on SL has also been conducted. Fischer et al. [28] studied 37 patients with idiopathic adrenal hyperplasia, and the mean period of SL treatment was 5.8 ± 0.7 years. Two of 37 (5.4%) patients were identified as having spontaneous remission, and one of the two patients was classified as having complete remission, with a normal aldosterone renin ratio (ARR), suppression test result, serum potassium level, and BP, while the other was classified as having partial remission due to persistent hypertension. Yoneda et al. [29] reported a case of spontaneous remission in a 41-year-old man with unilateral PA (UPA) after long-term treatment with SL. At the beginning of SL treatment, the patient’s ARR in the supine position was 603, and he had hypokalemia and a 5-mm left-sided adrenal tumor. Nine years later, SL treatment was discontinued, and the patient had a normal ARR, BP, and serum potassium level. These indicators have remained stable over the last 10 years, indicating that SL therapy might produce remission of UPA.

However, there were some different opinions in the study by Lucatello et al. [30], who thought that long-term treatment might play a role in the spontaneous remission of PA, but that the MRAs did not seem necessary. Of 34 recruited patients who had been classified as having PA for >3 years, with or without taking an aldosterone receptor blocker, 26 (76%) were no longer diagnosed with PA after a 1-month withdrawal of medical treatment, showing a significant association only with female sex, higher potassium levels, longer duration of hypertension, and follow-up. To explain this result, they suggested that aging was also mentioned in previous studies [26,27,28], which would cause a misunderstanding of the remission of PA because it was unclear which one should be the real reason for the remission: specific drug treatment or a biological spontaneous phenomenon.

## 4. Conclusions

In this review, we outlined the findings of basic and clinical studies regarding the remission phenomenon with MRAs in PA. These studies have elucidated the inhibitory effects of MRAs on steroidogenesis and aldosterone secretion, which might lead to spontaneous remission of PA after long-term treatment with MRAs. Further studies of PA remission are required.

## Figures and Tables

**Table 1 ijms-23-13821-t001:** Comparison of steroid hormone synthesis inhibition by MRAs.

References	Year	MRAs	Material	Steroid Hormone Synthesis & Production
[6]	1976	Spironolactone Canrenone Potassium carenoate	Adrenal Mitochondria from cows & humans	11β- ↓ 18- ↓
[7]	1977	Spironolactone	Human plasma	11β- ↓ 18- ↓
[10]	1979	Spironolactone	Adrenal and testicular microsomes from Sprague-Dawley rat & Guinea pig & dog	Cytochrome P-450 ↓ 17α- ↓ 21- ↓
[8]	1981	Spironolactone Canrenone	Adrenal mitochondria and microsomes from Guinea pig	11β- ↓ 21- ↓
[15]	1984	Spironolactone	Adrenal gland from rana ridibunda pallas frog	Aldosterone production ↓
[16]	2009	Spironolactone	Human adrenocorticoid H295R cells	Aldosterone production ↓ Cortisol production ↓
[17]	1990	Spironolactone 7α-thio-Spironolactone	Adrenocorticoid cells from Guinea pig	Cortisol production ↓
[12]	1991	Spironolactone 7α-thio-Spironolactone	Adrenocorticoid cells from Guinea pig	Cortisol production ↓ 17α- ↓
[9]	1978	Spironolactone Canrenone	Adrenal mitochondria and microsomes form Guinea pig	Cytochrome P-450 ↓ 11β- ↓ 21- ↓
[13]	1991	Spironolactone 7α-thio-Spironolactone	Adrenal microsomes from Guinea pig	Cytochrome P-450 ↓ 17α- ↓
[11]	1992	Spironolactone	Adrenal mitochondria and microsomes from Guinea pig	Cytochrome P-450 ↓ 17α- ↓ 21- ↓
[14]	1978	Spironolactone	Dog plasma	Concentration of testosterone ↓ estradiol ↓ and cortisol ↓

Abbreviations: 11β-, 11β-hydroxylase; 21-, 21-hydroxylase; 18-, 18-hydroxylase; 17α-, 17α-hydroxylase; MRAs, mineralocorticoid receptor antagonists; “↓”, decrease in steroid hormone synthesis activity, steroid hormone production or concentration.

**Table 2 ijms-23-13821-t002:** Analysis of spontaneous remission of primary aldosteronism after long-term treatment with mineralocorticoid receptor antagonists.

References	Year	Content
[18]	1981	Description of biochemical characteristics of SL-containing-cells
[19]	2006	Description of biochemical characteristics of SL-containing-cells
[20]	2016	Spironolactone and eplerenone could generally increase PAC
[21]	2013	Spironolactone and eplerenone could generally increase PAC
[22]	1974	Spironolactone could reduce PAC
[23]	2006	PAC was associated positively with the waist circumference in men
[24]	2002	PAC was associated negatively with high-density lipoprotein
[25]	2016	Chronic high dietary salt ingestion would suppress aldosterone secretion
[26]	2005	Potassium carenone could decrease ARR and might lead to a spontaneous remission after a long-term therapy with MRAs
[27]	1991	Spontaneous remission of PA after a long-term treatment with potassium carenone
[28]	2012	Spontaneous remission of PA after a long-term treatment with spironolactone
[29]	2012	Spontaneous remission of PA after a long-term treatment with spironolactone
[30]	2013	MRAs did not seem to be necessary for spontaneous remission of PA

Abbreviations: SL, spironolactone; PAC, plasma aldosterone concentration; ARR, aldosterone/renin ratio; MRAs, mineralocorticoid receptor antagonists; PA, primary aldosteronism.

## Data Availability

Not applicable.

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
