# Peer review of "Spontaneous Remission of Primary Aldosteronism with Mineralocorticoid Receptor Antagonist Therapy: A Review"

_ijms, 2022, doi:10.3390/ijms232213821_

Round 1

Reviewer 1 Report

In this review, Mai et al. described previous basic aspects of mineralocorticoid receptor antagonists. In addition, the authors presented retrospective data showing remission of primary aldosteronism after long-term treatment with spironolactone or eplerenone. This manuscript has major problems:

  • The manuscript has been written in a poor English.
  • The review part of this manuscript does not add any relevant aspect to what has been already published.
  • The original data demonstrating remission of primary aldosteronism after long-term treatment with spironolactone or eplerenone is potentially interesting, but should be written as an original article with more clinical and biochemical details.
  •  

Author Response

Reviewer 1: In this review, Mai et al. described previous basic aspects of mineralocorticoid receptor antagonists. In addition, the authors presented retrospective data showing remission of primary aldosteronism after long-term treatment with spironolactone or eplerenone. This manuscript has major problems:

The manuscript has been written in a poor English.

Response:

Thank you for pointing this out. This manuscript was proofread by Editage (www.editage.com) before submission. It was also proofread again after this revision.

The review part of this manuscript does not add any relevant aspect to what has been already published.

The original data demonstrating remission of primary aldosteronism after long-term treatment with spironolactone or eplerenone is potentially interesting, but should be written as an original article with more clinical and biochemical details.

Response:

Thank you for your suggestion. As indicated, we have excluded the unpublished study results from this manuscript, and we plan to report them as a separate original article.

Reviewer 2 Report

This is an interesting theme in the field of adrenal disease.

I have two major concerns.

In the aspect of clinical management, introduction of MRAs sometimes induce the upregulation of aldosterone concentration because of the effect as antagonist of MR. Authors focused on the inhibitory effect on the steroidogenesis of MRA, but they should mention this opposite phenomenon, which is often observed in patients with PA.

Did authors think that all of PA patients treated with MRAs show decreased steroidogenesis finally? Or, some patients show upregulation of PAC and patients with remission phenotype show downregulation of PAC?

Authors included unpublished data from 3.4. section. I think their original work should be reviewed and examined as original article, apart from this review.

Author Response

Reviewer 2: This is an interesting theme in the field of adrenal disease.

I have two major concerns.

In the aspect of clinical management, introduction of MRAs sometimes induce the upregulation of aldosterone concentration because of the effect as antagonist of MR. Authors focused on the inhibitory effect on the steroidogenesis of MRA, but they should mention this opposite phenomenon, which is often observed in patients with PA.

Response:

Thank you for your suggestion. As you kindly noted, the regulation of aldosterone concentration might be different in patients with PA. The regulation of PAC is associated with not only the medical treatments but also the metabolic syndrome, including the waist circumference in men [1], high-density lipoprotein [2], and salt intake [3]. The improvement in these factors will significantly decrease PAC and we should also pay more attention to them during MRA treatment. 

Did authors think that all of PA patients treated with MRAs show decreased steroidogenesis finally? Or, some patients show upregulation of PAC and patients with remission phenotype show downregulation of PAC?

Response:

Thank you for your question. We do not think that all PA patients will show decreased steroidogenesis after MRA treatment, as the actual effect may depend on their physical conditions. As we mentioned in this manuscript, “Although some of the positive results were meaningful and helpful for the study of SL, the difficulty between practical usage and experimental results is worth noting. Ye et al. [11] reported that the inhibitory effects of SL occurred at concentrations far higher than those needed to block mineralocorticoid receptors, up to 400 mg per day,” most PA patients mainly used normal dosage MRA treatment to control the hypertensive symptoms or heart failure, rather than to decrease steroidogenesis. Besides, some side effects of MRAs, such as hyperkalemia, are dose dependent [4], which could be another major reason why not all PA patents will get decreased steroidogenesis after MRA treatment.

As for the regulation of PAC, we [5] found that both spironolactone and eplerenone could generally increase PAC (135±57 to 213±90 pg/ml) in PA patients (n=54). A clinical study by Fourkiotis et al. [6], which compared the efficacies of eplerenone and spironolactone for PA patients (n=29), also supported this conclusion, as PAC rose from 58.3±9.0 to 159.3± 35.9 ng/l.

Authors included unpublished data from 3.4. section. I think their original work should be reviewed and examined as original article, apart from this review.

Response:

Thank you for your suggestion. As indicated, we have excluded the unpublished study results from this manuscript, and we plan to report them as a separate original article.

References:

  1. Bochud, M.; Nussberger, J.; Bovet, P.Maillard, M.R.; Elston, R.C.; Paccaud, F.; Shamlaye, C.; Burnier, M. Plasma aldosterone is independently associated with the metabolic syndrome. Hypertension, 2006, 48, 239-245, [doi:1161/01.HYP.0000231338.41548.fc].
  2. Goodfriend, T.L.; Egan, B.; Stepniakowski, K.; Ball, D.L. Relationships among plasma aldosterone, high-density lipoprotein cholesterol, and insulin in humans. Hypertension, 2002, 40, 892-296, [doi:1161/01.hyp.25.1.30].
  3. Calhoun, D.A.; Nishizaka, M.K.; Zaman, M.A.; Thakkar, R.B.; Weissmann, P. Hyperaldosteronism among Black and White subjects with resistant hypertension. Hypertension 2016, 68, 204-212, [doi:1161/01.hyp.0000040261.30455.b6].
  4. Struthers, A.; Krum, H.; Williams, G.H. A comparison of the aldosteron-blocking agents eplerenone and spironolactone. Cardiol. 2008, 31, 153-158, [doi:10.1002/clc.20324].
  5. Karashima, S.; Yoneda, T.; Kometani, M.; Ohe, M.; Mori, S.; Sawamura, T.; Furukawa, K.; Seta, T.; Yamagishi, M.; Takeda, Y. Comparison of eplerenone and spironolactone for the treatment of primary aldosteronism. Res. 2016, 39, 133–137, [doi:10.1038/hr.2015.129].
  6. Fourkiotis, V.; Vonend, O.; Diederich, S.; Fischer, E.; Lang, K.; Endres, S.; Beuschlein, F.; Willenberg, H.S.; Rump, L.C.; Allolio, B.; Reincke, M.; Quinkler, M.; Mephisto Study Group. Effectiveness of eplerenone or spironolactone treatment in preserving renal function in primary aldosteronism. J. Endocrinol. 2013, 168, 75-81, [doi:10.1530/EJE-12-0631].

Round 2

Reviewer 1 Report

The authors have addressed the raised points.

Reviewer 2 Report

Authors answered my questions appropriately.